# What is being transferred in transfer learning?

Behnam Neyshabur*
Google
neyshabur@google.com

Hanie Sedghi*
Google Brain
hsedghi@google.com

Chiyuan Zhang*
Google Brain
chiyuan@google.com

## Abstract

One desired capability for machines is the ability to transfer their knowledge of one domain to another where data is (usually) scarce. Despite ample adaptation of transfer learning in various deep learning applications, we yet do not understand what enables a successful transfer and which part of the network is responsible for that. In this paper, we provide new tools and analyses to address these fundamental questions. Through a series of analyses on transferring to block-shuffled images, we separate the effect of feature reuse from learning low-level statistics of data and show that some benefit of transfer learning comes from the latter. We present that when training from pre-trained weights, the model stays in the same basin in the loss landscape and different instances of such model are similar in feature space and close in parameter space.

## 1 Introduction

One desired capability of machines is to transfer their knowledge of a domain it is trained on (the source domain) to another domain (the target domain) where data is (usually) scarce or a fast training is needed. There has been a plethora of works on using this framework in different applications such as object detection [Girshick, 2015, Ren et al., 2015], image classification [Sun et al., 2017, Mahajan et al., 2018, Kolesnikov et al., 2019] and segmentation [Darrell et al., 2014, He et al., 2017], and various medical imaging tasks [Wang et al., 2017, Rajpurkar et al., 2017, De Fauw et al., 2018], to name a few. In some cases there is nontrivial difference in visual forms between the source and the target domain [Wang et al., 2017, Rajpurkar et al., 2017]. However, we yet do not understand what enables a successful transfer and which parts of the network are responsible for that. In this paper we address these fundamental questions.

To ensure that we are capturing a general phenomena, we look into target domains that are intrinsically different and diverse. We use CHEXPERT [Irvin et al., 2019] which is a medical imaging dataset of chest x-rays considering different diseases. We also consider DOMAINNET [Peng et al., 2019] datasets that are specifically designed to probe transfer learning in diverse domains. The domains range from real images to sketches, clipart and painting samples. See Figure 1 for sample images of the transfer learning tasks studied in this paper.

We use a series of analysis to answer the question of what is being transferred. First, we investigate feature-reuse by shuffling the data. In particular, we partition the image of the downstream tasks into equal sized blocks and shuffle the blocks randomly. The shuffling of blocks disrupts visual features in the images, especially with small block sizes (see Figure 1 for examples of shuffled images). We note the importance of feature re-use and we also see that it is not the only role player in successful transfer. This experiment shows that low-level statistics of the data that is not disturbed by shuffling the pixels also play a role in successful transfer (Section 3.1).

Figure 1: Sample images of dataset used for transfer learning downstream tasks. Left most: an example from CHEXPERT. The next three: an example from the DOMAINNET `real` dataset, the same image with random shuffling of $8 \times 8$ blocks and $1 \times 1$ blocks, respectively. The last three: examples from DOMAINNET `clipart` and `quickdraw`, and a $32 \times 32$ block-shuffled version of the `quickdraw` example.

Next, we compare the detailed behaviors of trained models. In particular, we investigate the agreements/disagreements between models that are trained from pre-training versus scratch. We note that two instances of models that are trained from pre-trained weights make similar mistakes. However, if we compare these two classes of models, they have fewer common mistakes. This suggests that two instances of models trained from pre-trained weights are more similar in feature space compared to ones trained from random initialization. To investigate this further, we look into feature similarity, measured by *centered kernel alignment* (CKA) [Kornblith et al., 2019a], at different modules[2] of the two model instances and observe that this is in fact the case. we also look into the $\ell_2$ distance between parameters of the models and note that two instances of models that are trained from pre-trained weights are much closer in $\ell_2$ distance compared to the ones trained from random initialization (Section 3.2).

We then investigate the loss landscape of models trained from pre-training and random initialization weights and observe that there is no performance barrier between the two instances of models trained from pre-trained weights, which suggests that the pre-trained weights guide the optimization to a flat basin of the loss landscape. On the other hand, barriers are clearly observed between the solutions from two instances trained from randomly initialized weights, even when *the same* random weights are used for initialization (Section 3.3).

The next question is, can we pinpoint where feature reuse is happening? As shown in [Zhang et al., 2019a] different modules of the network have different robustness to parameter perturbation. Chatterji et al. [2020] analyzed this phenomena and captured it under a module criticality measure. We extend this investigation to transfer learning with an improved definition of module criticality. Using this extended notion, we analyze criticality of different modules and observe that higher layers in the network have tighter valleys which confirms previous observations [Yosinski et al., 2014, Raghu et al., 2019] on features becoming more specialized as we go through the network and feature-reuse is happening in layers that are closer to the input. In addition, we observe that models that are trained from random initialization have a transition point in their valleys, which may be due to changing basins through training.

Finally, inspired by our findings about the basin of loss landscape, we look into different checkpoint in the training of the pre-trained model and show that one can start fine-tuning from the earlier checkpoints without losing accuracy in the target domain (see Section 3.5).

Our main contributions and takeaways are summarized below:

1. For a successful transfer both feature-reuse and low-level statistics of the data are important.

2. Models trained from pre-trained weights make similar mistakes on target domain, have similar features and are surprisingly close in $\ell_2$ distance in the parameter space. They are in the same basins of the loss landscape; while models trained from random initialization do not live in the same basin, make different mistakes, have different features and are farther away in $\ell_2$ distance in the parameter space.

3. Modules in the lower layers are in charge of general features and modules in higher layers are more sensitive to perturbation of their parameters.

Figure 2: Learning curves comparing random initialization (Rl-T) and finetuning from IMAGENET pre-trained weights(P-T). For CHEXPERT, finetune with base learning rate $0.1$ is not shown as it failed to converge.

4. One can start from earlier checkpoints of pre-trained model without losing accuracy of the fine-tuned model. The starting point of such phenomena depends on when the pre-train model enters its final basin.

## 2    Problem Formulation and Setup

In transfer learning, we train a model on the source domain and then try to modify it to give us good predictions on the target domain. In order to investigate transfer learning, we analyze networks in four different cases: the pre-trained network, the network at random initialization, the network that is fine-tuned on target domain after pre-training on source domain and the model that is trained on target domain from random initialization. Since we will be referring to these four cases frequently, we use the following notations. T: trained, P: Pre-trained, Rl: random initialization. Therefore we use the following abbreviations for the four models throughout the paper: Rl (random initialization), P (pre-trained model), Rl-T (model trained on target domain from random initialization), P-T (model trained/fine-tuned on target domain starting from pre-trained weights). We use IMAGENET [Deng et al., 2009] pre-training for its prevalence in the community and consider CHEXPERT [Irvin et al., 2019] and three sets from DOMAINNET [Peng et al., 2019] as downstream transfer learning tasks. See Appendix A for details about the experiment setup.

## 3    What is being transferred?

### 3.1    Role of feature reuse

Human visual system is compositional and hierarchical: neurons in the primary visual cortex (V1) respond to low level features like edges, while upper level neurons (e.g. the grandmother cell [Gross, 2002]) respond to complex semantic inputs. Modern convolutional neural networks trained on large scale visual data are shown to form similar feature hierarchies [Bau et al., 2017, Girshick et al., 2014]. The benefits of transfer learning are generally believed to come from reusing the pre-trained feature hierarchy. This is especially useful when the downstream tasks are too small or not diverse enough to learn good feature representations. However, this intuition cannot explain why in many successful applications of transfer learning, the target domain could be visually very dissimilar to the source domain. To characterize the role of feature reuse, we use a source (pre-train) domain containing natural images (IMAGENET), and a few target (downstream) domains with decreasing visual similarities from natural images: DOMAINNET real, DOMAINNET clipart, CHEXPERT and DOMAINNET quickdraw (see Figure 1). Comparing Rl-T to P-T in Figure 2, we observe largest performance boost on the real domain, which contains natural images that share similar visual features with IMAGENET. This confirms the intuition that feature reuse plays an important role in transfer learning. On the other hand, even for the most distant target domains such as CHEXPERT and quickdraw, we still observe performance boosts from transfer learning. Moreover, apart from the final performance, the optimization for P-T also converges much faster than Rl-T in all cases. This suggests additional benefits of pre-trained weights that are not directly coming from feature reuse.

To further verify the hypothesis, we create a series of modified downstream tasks which are increasingly distant from normal visual domains. In particular, we partition the image of the downstream tasks into equal sized blocks and shuffle the blocks randomly. The shuffling disrupts high level visual features in those images but keeps the low level statistics about the pixel values intact. The extreme case of block size $224 \times 224$ means no shuffling; in the other extreme case, all the pixels in the image are shuffled, making any of the learned visual features in pre-training completely useless. See Figure 1 for examples of block-shuffled images. Note that even for $1 \times 1$ blocks, all the RGB channels are moved around *together*. So we created a special case where pixels in each channel move

Figure 3: Effects on final performance (left: test accuracy) and optimization speed (right: average training accuracy over 100 finetune epochs) when the input images of the downstream tasks are block-shuffled. The x-axis shows the block sizes for shuffling. Block size '*' is similar to block size '1' except that pixel shuffling operates *across* all RGB channels. For each bar in the plots, the semi-transparent and solid bar correspond to initializing with pre-trained weights and random weights, respectively. The numbers on the top of the left pane show the relative accuracy drop: $100(A_{P\text{-}T} - A_{RI\text{-}T})/A_{P\text{-}T}\%$, where $A_{P\text{-}T}$ and $A_{RI\text{-}T}$ are test accuracy of models trained from pre-trained and random weights, respectively.

independently and could move to other channels. We then compare RI-T with P-T on those tasks[3]. The impacts on final performance and optimization speed with different block sizes are shown in Figure 3.

We observe that 1) The final performance drops for both RI-T and P-T as the block size decreases, indicating the increasing difficulty of the tasks. 2) The relative accuracy difference, measured as $(A_{P\text{-}T} - A_{RI\text{-}T})/A_{P\text{-}T}\%$, decreases with decreasing block size on both `real` and `clipart`, showing consistency with the intuition that decreasing feature reuse leads to diminishing benefits. 3) On the other hand, on `quickdraw`, the relative accuracy difference does not show a decreasing pattern as in the other two domains. This indicates that for `quickdraw`, where the input images are visually dissimilar to natural images, some other factors from the pre-trained weights are helping the downstream tasks. 4) The optimization speed on P-T is relatively stable, while on RI-T drops drastically with smaller block sizes. This suggests that benefits of transferred weights on optimization speed is independent from feature reuse.

We conclude that feature reuse plays a very important role in transfer learning, especially when the downstream task shares similar visual features with the pre-training domain. But there are other factors at play: in these experiments we change the size of the shuffled blocks all the way to 1 and even try shuffling the channels of the input, therefore, the only object that is preserved here is the set of all pixel values which can be treated as a histogram/distribution. We refer to those information as *low-level statistics*, to suggest that they are void of visual/semantic structural information. The low-level statistics lead to significant benefits of transfer learning, especially on optimization speed.

## 3.2 Opening up the model

**Investigating mistakes** In order to understand the difference between different models, we go beyond the accuracy values and look into the mistakes the models make on different samples of the data. We look into the *common mistakes* where both models classify the data point incorrectly and the *uncommon mistakes* (disagreements between models) where only one of them classifies the data point incorrectly and the other one does it correctly. We first compare the ratio of common and uncommon mistakes between two P-Ts, a P-T and a RI-T and two RI-Ts. We note a considerable number of uncommon mistakes between P-T and RI-T models while two P-Ts have strictly fewer uncommon mistakes. This trend is true for both CHEXPERT and DOMAINNET target domains.

We visualize the common and uncommon mistakes of each model on DOMAINNET, and observe that the data samples where P-T is incorrect and RI-T is correct mostly include ambiguous examples; whereas the data samples where P-T is correct, RI-T is incorrect include a lot of easy samples too. This complies with the intuition that since P-T has stronger prior, it harder to adapt to the target domain. Moreover, when we repeat the experiment for another instance of P-T, the mistaken examples between two instances of P-T are very similar, see Appendix B.2 for visualizations of such data points. This may suggest that two P-T's are more similar in the feature space compared to

two RI-T's and P-T vs RI-T. We investigate this idea further by looking into similarity of the two networks in the feature space.

**Feature Similarity**   We use the *centered kernel alignment* (CKA) [Kornblith et al., 2019b] as a measure of similarity between two output features in a layer of a network architecture given two instances of such network. CKA [Kornblith et al., 2019b] is the latest work on estimating feature similarity with superior performance over earlier works. The results are shown in Table 1. We observe that two instances of P-T are highly similar across different layers. This is also the case when we look into similarity of P-T and P. However, between P-T and RI-T instance or two RI-T instances, the similarity is very low. Note that the feature similarity is much stronger in the penultimate layer than any earlier layers both between P-T and RI-T instance and two RI-T instances, however, still an order of magnitude smaller than similarity between two P-T layers.

These experiments show that the initialization point, whether pre-trained or random, drastically impacts feature similarity, and although both networks are showing high accuracy, they are not that similar in the feature space. This emphasizes on role of feature reuse and that two P-T are reusing the same features.

Table 1: Feature similarity for different layers of ResNet-50, target domain CHEXPERT

| models/layer | conv1 | layer 1 | layer 2 | layer 3 | layer 4 |
|---|---|---|---|---|---|
| P-T & P | 0.6225 | 0.4592 | 0.2896 | 0.1877 | 0.0453 |
| P-T & P-T | 0.6710 | 0.8230 | 0.6052 | 0.4089 | 0.1628 |
| P-T & RI-T | 0.0036 | 0.0011 | 0.0022 | 0.0003 | 0.0808 |
| RI-T & RI-T | 0.0016 | 0.0088 | 0.0004 | 0.0004 | 0.0424 |

**Distance in parameter space**   In addition to feature similarity, we look into the distance between two models in the parameter space. More specifically, we measure the $\ell_2$ distance between 2 P-Ts and 2 RI-Ts, both per module and for the entire network. Interestingly, RI-Ts are farther from each other compared to two P-Ts, (see Table 2) and this trend can be seen in individual modules too (see Figure 10, 11, Table 8 in the Appendix for more comparisons). Moreover the distance between modules increases as we move towards higher layers in the network.

### 3.3   Performance barriers and basins in the loss landscape

A commonly used criterion for better generalization performance is the flatness of the basin of the loss landscape near the final solution. In a flat basin, the weights could be locally perturbed without hurting the performance, while in a narrow basin, moving away from the minimizer would quickly hit a *barrier*, indicated by a sudden increase in the loss.

To explore the loss landscape of P-T and RI-T, we use the following procedure to identify potential performance barriers. Let $\Theta$ and $\tilde{\Theta}$ be all the weights from two different checkpoints. We evaluate a series of models along the *linear* interpolation of the two weights: $\{\Theta_\lambda = (1-\lambda)\Theta + \lambda\tilde{\Theta} : \lambda \in [0,1]\}$. It has been observed in the literature that any two minimizers of a deep network can be connected via a *non-linear* low-loss path Garipov et al. [2018], Draxler et al. [2018], Fort and Jastrzebski [2019]. In contrast, due to the non-linear and compositional structure of neural networks, the *linear* combination of the weights of two good performing models does not necessarily define a well behaved model, thus performance barriers are generally expected along the linear interpolation path. However, in the case when the two solutions belong to the same flat basin of the loss landscape, the linear interpolation remains in the basin. As a result, a performance barrier is absent. Moreover, interpolating two random solutions from the same basin could generally produce solutions closer to the center of the basin, which potentially have better generalization performance than the end points.

Table 2: Features $\ell_2$ distance between two P-T and two RI-T for different target domains

| domain/model | 2 P-T | 2 RI-T | P-T & P | RI-T & P |
|---|---|---|---|---|
| CHEXPERT | 200.12 | 255.34 | 237.31 | 598.19 |
| clipart | 178.07 | 822.43 | 157.19 | 811.87 |
| quickdraw | 218.52 | 776.76 | 195.44 | 785.22 |
| real | 193.45 | 815.08 | 164.83 | 796.80 |

Figure 4: Performance barrier between different solutions. The left and middle panes show performance barrier measured by test accuracy on DOMAINNET `real` and `quickdraw`, respectively. The right pane shows the performance barrier measured by test AUC on CHEXPERT. In the legend, 'randinit*' and 'randinitT' means the best and final checkpoint in a RI-T training trajectory. Similarly, 'finetune*' and 'finetuneT' are for P-T. Since there is no overfitting from overtraining on DOMAINNET, we only show results for final checkpoints.

The word "basin" is often used loosely in the literature to refer to areas in the parameter space where the loss function has relatively low values. Since prior work showed that non-linear low-loss path could be found to connect any pair of solutions, we focus on convex hull and linear interpolation in order to avoid trivial connectivity results. In particular, we require that for most points on the basin, their convex combination is on the basin as well. This extra constraint would allow us to have multiple basins that may or may not be connected though a low-loss (nonlinear) path. We formalize this notion as follows:

**Definition 3.1.** *Given a loss function $\ell : \mathbb{R}^n \to \mathbb{R}^+$ and a closed convex set $S \subset \mathbb{R}^n$, we say that $S$ is a $(\epsilon, \delta)$-basin for $\ell$ if and only if $S$ has all following properties:*

1. *Let $U_S$ be the uniform distribution over set $S$ and $\mu_{S,\ell}$ be the expected value of the loss $\ell$ on samples generated from $U_S$. Then,*

$$\mathbb{E}_{\mathbf{w} \sim U_S}[|\ell(\mathbf{w}) - \mu_{S,\ell}|] \leq \epsilon \tag{1}$$

2. *For any two points $w_1, w_2 \in S$, let $f(w_1, w_2) = w_1 + \tilde{\alpha}(w_2 - w_1)$, where $\tilde{\alpha} = \max\{\alpha | w_1 + \alpha(w_2 - w_1) \in S\}$. Then,*

$$\mathbb{E}_{\mathbf{w_1}, \mathbf{w_2} \sim U_S, \nu \sim \mathcal{N}(0, (\delta^2/n)I_n)}[\ell(f(\mathbf{w_1}, \mathbf{w_2}) + \nu) - \mu_{S,\ell}] \geq 2\epsilon \tag{2}$$

3. *Let $\kappa(\mathbf{w_1}, \mathbf{w_2}, \nu) = f(\mathbf{w_1}, \mathbf{w_2}) + \frac{\nu}{\|f(\mathbf{w_1}, \mathbf{w_2}) - \mathbf{w_1}\|_2}(f(\mathbf{w_1}, \mathbf{w_2}) - \mathbf{w_1})$. Then,*

$$\mathbb{E}_{\mathbf{w_1}, \mathbf{w_2} \sim U_S, \nu \sim \mathcal{N}(0, \delta^2)}[\ell(\kappa(\mathbf{w_1}, \mathbf{w_2}, |\nu|)) - \mu_{S,\ell}] \geq 2\epsilon \tag{3}$$

Based on the above definition, there are three requirements for a convex set to be a basin. The first requirement is that for most points on the basin, their loss should be close to the expected value of the loss in the basin. This notion is very similar to requiring the loss to have low variance for points on the basin[4]. The last two requirements ensure that the loss of points in the vicinity of the basin is higher than the expected loss on the basin. In particular, the second requirement does that by adding Gaussian noise to the points in the basin and requiring the loss to be higher than the expected loss in the basin. The third requirement does something similar along the subspaces spanned by extrapolating the points in the basin. That is, if one exits the basin by extrapolating two points on the basin, the loss should increase.

In Figure 4 we show interpolation results on DOMAINNET `real`, `quickdraw`, and CHEXPERT. Generally, we observe no performance barrier between the P-T solutions from two random runs, which suggests that the pre-trained weights guide the optimization to a flat basin of the loss landscape. Moreover, there are models along the linear interpolation that performs slightly better than the two end points, which is again consistent with our intuition. On the other hand, barriers are clearly observed between the solutions from two RI-T runs (even if we use the same random initialization values).

Figure 6: Module Criticality plots for Conv1 module. x-axis shows the distance between initial and optimal $\theta$, where $x = 0$ maps to initial value of $\theta$. y-axis shows the variance of Gaussian noise added to $\theta$. The four subplots refer to the four paths one can use to measure criticality. All of which provide good insight into this phenomenon. Heat map is used so that the colors reflect the value of the measure under consideration.

On CHEXPERT, because the models start to overfit after certain training epochs (see Figure 2), we examine both the best and final checkpoints along the training trajectory, and both show similar phenomena. Interestingly, the interpolation between the best and the final checkpoints (from the same training trajectory) also shows a (small) barrier for RI-T (but not for P-T).

Starting with the two P-T solutions, we extrapolated beyond their connecting intervals to find the basin boundary, and calculated the parameters according to Definition 3.1. We found that each pair of P-T solutions live in a $(0.0038, 49.14)$-basin, $(0.0054, 98.28)$-basin and $(0.0034, 49.14)$-basin for real, clipart and quickdraw, respectively. Their $\mu_{S,\ell}$ values are $0.2111$, $0.2152$ and $0.3294$, respectively, where $\ell$ measures the test error rate. On the other hand, pairs of RI-T solutions do not live in the same $(\epsilon, \delta)$-basin for reasonable $\epsilon$ thresholds.

We have done a variety of additional experiments on analysing the performance barriers for different domains, cross-domains, extrapolation and training on combined domains that can be found in the Appendix.

## 3.4 Module Criticality

It has been observed that different layers of the network show different robustness to perturbation of their weight values [Zhang et al., 2019a]. Zhang et al. [2019a] did the following experiment: consider a trained network, take one of the modules and rewind its value back to its initial value while keeping the weight value of all other modules fixed at trained values. They noted that for some modules, which they called *critical*, the performance of the model drops significantly after rewinding, while for others the performance is not impacted. Chatterji et al. [2020] investigated this further and formulated it by a notion of module criticality. Module criticality is a

Figure 5: Module criticality measured on CHEXPERT.

measure that can be calculated for each module and captures how critical each module is. It is defined in [Chatterji et al., 2020] by looking at width of the valleys that connect the final and initial value of weight matrix of a module, while keeping all the other modules at their final trained value.

**Definition 3.2** (Module Criticality [Chatterji et al., 2020]). *Given an $\epsilon > 0$ and network $f_\Theta$, we define the module criticality for module $i$ as follows:*

$$\mu_{i,\epsilon}(f_\Theta) = \min_{0 \le \alpha_i, \sigma_i \le 1} \left\{ \frac{\alpha_i^2 \left\| \theta_i^F - \theta_i^0 \right\|_{\mathrm{Fr}}^2}{\sigma_i^2} : \mathbb{E}_{u \sim \mathcal{N}(0,\sigma_i^2)}[\mathcal{L}_S(f_{\theta_i^\alpha + u, \Theta_{-i}^F})] \le \epsilon \right\}, \qquad (4)$$

*where $\theta_i^\alpha = (1 - \alpha)\theta_i^0 + \alpha\theta_i^F, \alpha \in [0,1]$ is a point on convex combination path between the final and initial value of the weight matrix $\theta_i$ and we add Gaussian perturbation $u \sim \mathcal{N}(0, \sigma_i^2)$ to each point.*

Chatterji et al. [2020] showed that module criticality captures the role of the module in the generalization performance of the whole architecture and can be used as a measure of capacity of the module and predict the generalization performance. In this paper, we extend the definition of module criticality by looking at both *direct path* that linearly connect the initial and final value of the module and the *optimization path* generated by an optimizer from initialization to the final solution

Figure 7: Comparing the final performance (left: test accuracy after finetuning) and optimization speed (right: average training accuracy over 100 finetune epochs) of transfer learning from different pre-training checkpoints. The purple shaded area indicates the top-1 accuracy of the IMAGENET pre-training task at each of the checkpoints.

(checkpoints during training). We also look into the final value of a weight matrix in addition to its optimal value during training as the start point of the path for investigating criticality, i.e., instead of $\theta_i^F$, we investigate $\theta_i^{\text{opt}}$ which is the checkpoint where the network has the best validation accuracy during training. Moreover, we ensure that the noise is proportional to the Frobenius norm of weight matrix. Similar to [Chatterji et al., 2020], we define the network criticality as the sum of the module criticality over modules of the network. Note that we can readily prove the same relationship between this definition to generalization performance of the network as the one in [Chatterji et al., 2020] since the proof does not depend on the start point of the path.

In Figure 5 we analyze criticality of different modules in the same way as [Zhang et al., 2019a]. We note a similar pattern as observed in the supervised case. The only difference is that the 'FC' layer becomes critical for P-T model, which is expected. Next, we investigate crititcality of different modules with our extended definition along with original definition. We note that both optimization and direct paths provide interesting insights into the criticality of the modules. We find that the optimal value of weight is a better starting point for this analysis compared to its final value. Figure 6 shows this analysis for 'Conv1' module, which as shown in Figure 5 is a critical module. In this Figure we only look at the plots when measuring performance on training data. As shown in Figure 30 (Appendix B.9), we can see the same trend when looking at the test performance or generalization gap.

As we move from the input towards the output, we see tighter valleys, i.e., modules become more critical (See the Supplementary file for criticality plots of different layers). This is in agreement with observation of [Yosinski et al., 2014, Raghu et al., 2019] that lower layers are in charge of more general features while higher layers have features that are more specialized for the target domain. Moreover, For modules of the RI-T model, we notice more sensitivity and a transition point in the path between optimal and initial point, where the valley becomes tighter and wider as we move away from this point. Whether this is related to the module moving to another basin is unclear.

### 3.5 Which pre-trained checkpoint is most useful for transfer learning?

We compare the benefits of transfer learning by initializing the pre-trained weights from different checkpoints on the pre-training optimization path. Figure 7 shows the final performance and optimization speed when finetuning from different pre-training checkpoints. Overall, the benefits from pre-training increase as the checkpoint index increases. Closer inspection reveals the following observations: 1) in pre-training, big performance boosts are observed at epoch 30 and 60 where the learning rate decays. However, initializing from checkpoints 29, 30, 31 (and similarly 59, 60, 61) does not show significantly different impact. On the other hand, especially for final performance of `real` and `clipart`, significant improvements are observed when we start from the checkpoints where the pre-training performance has been plateauing (i.e. checkpoint 29 and 59). This shows that the pre-training performance is not always a faithful indicator of the effectiveness of the pre-trained weights for transfer learning. 2) `quickdraw` sees much smaller benefit on final performance from pre-training and quickly plateaus at checkpoint 10, while `real` and `clipart` continuously see noticeable performance improvements until checkpoint 60. On the other hand, all three tasks get significant benefits on optimization speed improvements as the checkpoint index increases. 3) The optimization speedups start to plateau at checkpoint 10, while (for `real` and `clipart`) the final performance boosts continue to increase.

In summary, we observe independence between the improvements on optimization speed and final performance. Moreover, this is in line with the loss landscape observations in Section 3.3. Earlier checkpoints in pre-training are out of basin of the converged model and at some point during training we enter the basin (which is the same for pre-train and fine-tune models as we saw in Section 3.3). This also explains the plateau of performance after some checkpoints. Therefore, we can start from earlier checkpoints in pre-training.

## 4 Related work

Recent work has looked into whether transfer learning is always successful [He et al., 2018, Kornblith et al., 2018, Ngiam et al., 2018, Huh et al., 2016, Geirhos et al., 2019]. For example, Kornblith et al. [2018] illustrates that pretrained features may be less general than previously thought and He et al. [2018] show that transfer (even between similar tasks) does not necessarily result in performance improvements. Kolesnikov et al. [2019] propose a heuristic for setting the hyperparameters for transfer. Yosinski et al. [2014] show that in visual domain features from lower layers are more general and as we move more towards higher layers the features become more specific. Raghu et al. [2020] evaluated the role of feature reusing in meta-learning. Directly related to our work is [Raghu et al., 2019], where they investigate transfer learning from pre-trained IMAGENET model to medical domain and note the role of model size on transfer performance, as well as the role of feature independent aspects such as weight scaling. In this paper, we go beyond their angles and propose an extensively complementary analysis. We dissect the role of different modules from various angles. Moreover, we provide insights into what is being transferred. We also extend the understanding of role of feature reuse and other parameters at play for successful transfer.

Transfer learning is also a key components in recent breakthroughs in natural language processing, as pre-trained representations from task-agnostic transformer language models turned out to work extremely well on various downstream tasks [Devlin et al., 2018, Raffel et al., 2019, Yang et al., 2019, Liu et al., 2019, Brown et al., 2020], therefore extensively analyzed [Hendrycks et al., 2020, Tamkin et al., 2020]. Transfer learning in text domains is charasteristically different from transfer learning in visual domains due to the nature of the data, utility of untrained representations, pre-training techniques and the amount of distribution shifts between pre-training and downstream tasks. In this paper, we focus on studying transfer learning in visual domain.

The flatness of the basin of the loss landscape near the final solution is studied as an indicator of the generalization performance of neural network models [Keskar et al., 2016, Jiang et al., 2019]. It is found that a *nonlinear* connecting path could be found between any pair of basins corresponding to different solutions [Garipov et al., 2018, Draxler et al., 2018, Fort and Jastrzebski, 2019]. In this paper, we study *linear* path connectivity between solutions to investigate the connectivity in the parameter space.

## 5 Conclusion and future work

In this paper we shed some light on what is being transferred in transfer learning and which parts of the network are at play. We investigated the role of feature reuse through shuffling the blocks of input and showed that when trained from pre-trained weights initialization, the network stays in the same basin of the solution, features are similar and models are close in the $\ell_2$ distance in parameter space. We also confirmed that lower layers are in charge of more general features. Our findings on basin in the loss landscape can be used to improve ensemble methods. Our observation of low-level data statistics improving training speed could lead to better network initialization methods. Using these findings to improve transfer learning is of interest for future work. More specifically, we plan to look into initialization with minimum information from pre-trained model while staying in the same basin and whether this improves performance. For example, one can use top singular values and directions for each module for initialization and investigate if this suffices for good transfer, or ensure initialization at the same basin but adding randomization to enhance diversity and improve generalization. Investigating the implications of these findings for parallel training and optimization is also of interest. taking model average of models in the same basin does not disturb the performance.

## Broader Impact

Transfer learning requires using a model that is trained on different data and adapt it to new data distribution. The difference in data distribution brings risk. Proper transfer moves the model towards distribution of data in the target domain. Our goal is to understand transfer learning to improve its performance and reduce the risk. This work is a foundational analysis.

## Acknowledgments and Disclosure of Funding

We would like to thank Samy Bengio and Maithra Raghu for valuable conversations. This work was funded by Google.

## Footnotes

*Equal contribution. Authors ordered randomly.

[2]A module is a node in the computation graph that has incoming edges from other modules and outgoing edges to other nodes and performs a linear transformation on its inputs. For layered architectures such as VGG, each layer is a module. For ResNets, a module can be a *residual block*, containing multiple layers and a skip connection.

[3]We disable data augmentation (random crop and left-right flip) during finetuning because they no longer make sense for block-shuffled images and make optimization very unstable.

[4]Here the term that captures the difference has power one as opposed to variance where the power is two.

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
