[Supplementary Material 1]

# Appendix 'What is being transferred in transfer learning?'

# Appendix

## Table of Contents

## Appendix A: Experiment Setup

We use IMAGENET [Deng et al., 2009] pre-training for its prevalence in the community and consider CHEXPERT [Irvin et al., 2019] and three sets from DOMAINNET [Peng et al., 2019] as downstream transfer learning tasks. CHEXPERT is a medical imaging dataset which consists of chest x-ray images. We resized the x-ray images to to $224 \times 224$, and set up the learning task to diagnose 5 different thoracic pathologies: atelectasis, cardiomegaly, consolidation, edema and pleural effusion. DOMAINNET [Peng et al., 2019] is a dataset of common objects in six different domain. All domains include 345 categories (classes) of objects such as Bracelet, plane, bird and cello . The domains include `clipart`: collection of clipart images; `real`: photos and real world images; `sketch`: sketches of specific objects; `infograph`: infographic images with specific object; `painting` artistic depictions of objects in the form of paintings and `quickdraw`: drawings of the worldwide players of game "Quick Draw!"[5]. In our experiments we use three domains: `real`, `clipart` and `quickdraw`.

The CHEXPERT dataset default split contains a training set of 200k images and a tiny validation set that contains only 200 studies. The drastic size difference is because the training set is constructed using an algorithmic labeler based on the free text radiology reports while the validation set is manually labeled by board-certified radiologists. To avoid high variance in the studies due to tiny dataset size and label distribution shift, we do not use the default partition. Instead, we sample two separate sets of 50k examples from the full training set and use them as the train and test set, respectively. Different domains in DOMAINNET contain different number of training examples, ranging from 50k to 170k. To keep the setting consistent with CHEXPERT, We sample 50k subsets from training and test sets for each of the domains.

Table 3: Common and uncommon mistakes between RI-T, P-T, CHEXPERT

|         | P-T acc | RI-T acc | g1   | g2   | Common mistakes | r1     | r2      |
|---------|---------|----------|------|------|-----------------|--------|---------|
| Class 1 | 0.8830  | 0.8734   | 645  | 832  | 3597            | 0.1878 | 0.1520  |
| Class 2 | 0.7490  | 0.7296   | 2361 | 2318 | 4047            | 0.3641 | 0.3684  |
| Class 3 | 0.9256  | 0.9095   | 71   | 207  | 3009            | 0.0643 | 0.02305 |
| Class 4 | 0.6811  | 0.6678   | 2250 | 2460 | 8835            | 0.2177 | 0.2029  |
| Class 5 | 0.7634  | 0.7374   | 3446 | 2160 | 4200            | 0.3396 | 0.4506  |

Table 4: Common and uncommon mistakes between two instances of RI-T, CHEXPERT

|         | P-T acc | RI-T acc | g1   | g2   | Common mistakes | r1     | r2     |
|---------|---------|----------|------|------|-----------------|--------|--------|
| Class 1 | 0.8734  | 0.8654   | 704  | 773  | 3469            | 0.1822 | 0.1687 |
| Class 2 | 0.7292  | 0.7399   | 3139 | 1499 | 4909            | 0.2339 | 0.3900 |
| Class 3 | 0.9095  | 0.9170   | 195  | 102  | 2978            | 0.0331 | 0.0614 |
| Class 4 | 0.6678  | 0.6574   | 2070 | 2588 | 8497            | 0.2334 | 0.1958 |
| Class 5 | 0.7374  | 0.7326   | 2001 | 3136 | 4510            | 0.4101 | 0.3073 |

For all our training and transfer experiments we use ResNet-50 [He et al., 2016] with fixup initialization [Zhang et al., 2019b] which eliminate batch normalization layers from the ResNet architecture. We initialize the final linear classifier layer with uniform random values instead of using zeros from the fixup initialization.

We run each CHEXPERT training jobs with two NVidia V100 GPUs, using batch size 256. We use the SGD optimizer with momentum 0.9, weight decay 0.0001, and constant learning rate scheduling. We also tried piece-wise constant learning rate scheduling. However, we found that RI-T struggles to continue learning as learning rate decays. We train P-T for 200 epochs and RI-T for 400 epochs, long enough for both training scenarios to converge to the final (overfitted) solutions. It takes $\sim 90$ seconds to train one epoch. We also run full evaluation on both the training and test sets each epoch, which takes $\sim 40$ seconds each.

We run each DOMAINNET training jobs with single NVidia V100 GPU, using batch size 32. We use the SGD optimizer with momentum 0.9, weight decay 0.0001, and piecewise constant learning rate scheduling that decays the learning rate by a factor of 10 at epoch 30, 60, and 90, respectively. We run the training for 100 epochs. In each epoch, training takes around 2 minutes and 45 seconds, and evaluating on both the training and test set takes around 70 seconds each.

## Appendix B: Additional Figures and Tables

### B.1  Discussions of learning curves

Figure 2 shows the learning curves for P-T, RI-T models with different learning rates. In particular, we show base learning rate 0.1 and 0.02 for both cases on CHEXPERT, DOMAINNET real, clipart and quickdraw, respectively.

We observe that P-T generally prefer smaller learning rate, while RI-T generally benefit more from larger learning rate. With large learning rate, P-T failed to converge on CHEXPERT, and significantly under-performed the smaller learning rate counterpart on DOMAINNET quickdraw. On DOMAINNET real and clipart, the gap is smaller but smaller learning rate training is still better. On the other hand, with larger learning rate, RI-T significantly outperform the smaller learning rate counterpart on all the three DOMAINNET datasets. On CHEXPERT, although the optimal and final performance are similar for both small and large learning rate, the large learning version converges faster.

On all the four datasets, P-T outperforms RI-T, in both optimization speed and test performance. Note that we subsample all the four dataset to $50,000$ training examples of $224 \times 224$ images, yet severe overfitting is observed only on CHEXPERT. This suggests that the issue of overfitting is not only governed by the problem size, but also by other more complicated factors such as the nature of input images. Intuitively, the chest X-ray images from CHEXPERT are less diverse than the images from DOMAINNET.

Table 5: Common and uncommon mistakes between two instances of P-T, CHEXPERT

|  | P-T acc | RI-T acc | g1 | g2 | Common mistakes | r1 | r2 |
|---|---|---|---|---|---|---|---|
| Class 1 | 0.8830 | 0.8828 | 456 | 697 | 3732 | 0.1573 | 0.1088 |
| Class 2 | 0.7490 | 0.7611 | 2375 | 1369 | 4996 | 0.2150 | 0.3222 |
| Class 3 | 0.9256 | 0.9208 | 63 | 139 | 3077 | 0.0432 | 0.0200 |
| Class 4 | 0.6811 | 0.6825 | 2030 | 1879 | 9416 | 0.1663 | 0.1773 |
| Class 5 | 0.76344 | 0.76088 | 1687 | 2398 | 3962 | 0.3770 | 0.2986 |

Table 6: Common and uncommon mistakes between P-T, RI-T, `clipart`

|  | g1 | g2 | Common mistakes | r1 | r2 |
|---|---|---|---|---|---|
| P-T, RI-T | 521 | 2940 | 3263 | 0.1376 | 0.4739 |
| RI-T, RI-T | 787 | 844 | 5359 | 0.1280 | 0.1360 |
| P-T, P-T | 619 | 584 | 3165 | 0.1635 | 0.1558 |

## B.2 Common and uncommon mistakes

In order to look into common and uncommon mistakes, we compare two models at a time. We look into all combinations, i.e., compare RI-T, P-T, two instances of RI-T and two instances of P-T. Tables 3, 4, 5 show this analysis for CHEXPERT and Table 6 shows the analysis for `clipart`. For CHEXPERT we do a per class analysis first. Since CHEXPERT looks into five different diseases, we have five binary classification tasks. In each classification setting, we look into accuracy of each model. g1, g2 refer to the number of data samples where only the first model is classifying correctly and only the second model is classifying correctly, respectively. We also look into the number of common mistakes. r1, r2 refer to ratio of uncommon to all mistakes for the first and second model respectively.

Another interesting point about DOMAINNET is that classes are not balanced. Therefore, we investigate the correlation of P-T and RI-T accuracy with class size. The results are shown in Figure 8 for `clipart`. Other target domains from DOMAINNET show similar trends. We also compute the Pearson correlation coefficient [Pearson, 1895] between per class accuracy's and class sizes for these models and they are P-T $(0.36983, 1.26e - 12)$, RI-T $(0.32880, 3.84e - 10)$. Note that overall top-1 accuracy of P-T, RI-T is $74.32, 57.91$ respectively. In summary, P-T has higher accuracy overall, higher accuracy per class and it's per-class accuracy is more correlated with class size compared to RI-T.

## B.3 Feature similarity and different distances

Table 7 shows feature similarity using CKA Kornblith et al. [2019a] for output of different layers of ResNet-50 when the target domain is `clipart`. In Section 3.2 in the main text we showed a similar

Figure 8: Accuracy of P-T and RI-T vs class size for `clipart`

Figure 9: Uncommon Mistakes RI-T and P-T, top row shows samples of the images RI-T classifies incorrectly while P-T classifies them correctly. bottom row shows samples of the images P-T classifies incorrectly while RI-T classifies them correctly. Classes from left to right are: barn, apple, backpack, angel.

Table 7: Feature similarity for different layers of ResNet-50, target domain `clipart`

| models/layer | conv1 | layer 1 | layer 2 | layer 3 | layer 4 |
|---|---|---|---|---|---|
| P-T & P | 0.3328 | 0.2456 | 0.1877 | 0.1504 | 0.5011 |
| P-T & P-T | 0.4333 | 0.1943 | 0.4297 | 0.2578 | 0.1167 |
| P-T & RI-T | 0.0151 | 0.0028 | 0.0013 | 0.008 | 0.0014 |
| RI-T & RI-T | 0.0033 | 0.0032 | 0.0088 | 0.0033 | 0.0012 |

table for target domain CHEXPERT. We observe a similar trend when we calculate these numbers for `quickdraw` and `real` target domain as well.

Figure 10 depicts $\ell_2$ distance between modules of two instances of P-T and two instances of RI-T for both CHEXPERT and `clipart` target domain. Table 2 shows the overall distance between the two networks. We note that P-T's are closer in $\ell_2$ parameter domain compared to RI-Ts. Drawing this plot for `real`, `quickdraw` leads to the same conclusion.

We also looked into distance to initialization per module, and overall distance to initialization for P-T, RI-T for different target domains in Figure 11 and Table 8.

## B.4 Additional plots for performance barriers

Figure 12 shows the performance barrier plots measured with on all the three DOMAINNET datasets. Figure 13 show results measured with the cross entropy loss. On both plots, we observe performance barrier between two RI-T solutions, but not between two P-T solutions.

Table 8: Distance to initialization between P-T and RI-T for different target domains

| domain/model | P-T | RI-T |
|---|---|---|
| CHEXPERT | 4984 | 4174 |
| clipart | 5668 | 23249 |
| quickdraw | 7501 | 24713 |
| real | 5796 | 24394 |

(a) CHEXPERT           (b) clipart

Figure 10: Feature $\ell_2$ distance per module

(a) CHEXPERT           (b) clipart

Figure 11: Distance to Initialization per module

Figure 14 and Figure 15 show the performance barrier plots of CHEXPERT measured by AUC and loss, respectively. The two panes in each of the figures show the two cases where RI-T is trained with learning rate $0.1$ and $0.02$, respectively. Note that on CHEXPERT the models overfit afer a certain number of training epochs and the final performances are worse than the optimal performances along the training trajectory. So we show more interpolation pairs than in the case of DOMAINNET. Those plots are largely consistent with our previous observations. One interesting observation is that while the final performance of P-T is better than RI-T when measured with (test) AUC, the former also has higher (test) loss than the latter.

## B.5    Performance barrier experiments with identical initialization for RI-T

In the experiments of comparing the performance barrier interpolating the weights of two RI-T models vs. interpolating the weights of two P-T models, the two P-T models are initialized from *the same* pre-trained weights, while the two RI-T models are initialized from independently sampled (therefore *different*) random weights. In this section, we consider interpolating two RI-T models that are trained from *identical* (random) initial weights. The results are shown in Figure 16 and Figure 17. Comparing with their counterparts in Figure 12 and Figure 13, respectively, we found that the barriers

Figure 12: Performance barrier of `real`, `clipart`, `quickdraw`, respectively, measured by test accuracy.

Figure 13: Loss barrier of `real`, `clipart`, `quickdraw`, respectively, measured by the cross entropy loss.

Figure 14: Performance barrier on CHEXPERT. Left: RI-T is using base learning rate 0.1; Right: RI-T is using base learning rate 0.02.

Figure 15: Loss barrier on CHEXPERT. Left: RI-T is using base learning rate $0.1$; Right: RI-T is using base learning rate $0.02$.

Figure 16: Performance barrier of `real`, `clipart`, `quickdraw`, respectively, measured by test accuracy. Like P-T, the two RI-T models are initialized from *the same* (random) weights. This figure can be compared with Figure 12.

between two RI-T models become slightly smaller when the initial weights are the same. However, significant barriers still exist when comparing with the interpolation between two P-T models.

## B.6 Performance barrier plots with extrapolation

In order to estimate the boundary of the basin according to Definition 3.1, we extend the interpolation coefficients from $[0, 1]$ to extrapolation in $[-1, 2]$. The results are shown in Figure 18. We can see that in this 1D subspace, the two P-T solutions are close to the boundary of the basin, while RI-T solutions do not live in the same basin due to the barriers on the interpolating linear paths.

## B.7 Cross-domain weight interpolation on DOMAINNET

Because all the domains in DOMAINNET have the same target classes, we are able to directly apply a model trained on one domain to a different domain and compute the test performance. Moreover, we can also interpolate the weights between models that are trained on different domains. In particular, we tested the following scenarios:

Figure 17: Performance barrier of `real`, `clipart`, `quickdraw`, respectively, measured by cross entropy loss. Like P-T, the two RI-T models are initialized from *the same* (random) weights. This figure can be compared with Figure 13.

Figure 18: Performance barrier of `real`, `clipart`, `quickdraw`, respectively, measured by test accuracy. The linear combination of weights are extrapolated beyond $[0, 1]$ (to $[-1, 2]$).

| Results | Evaluated on | Training data for Model 1 | Training data for Model 2 |
|---------|-------------|---------------------------|---------------------------|
| Figure 19 | clipart | clipart | real |
| Figure 20 | clipart | quickdraw | real |
| Figure 21 | real | quickdraw | clipart |
| Figure 22 | real | real | clipart |
| Figure 23 | real | real | quickdraw |

It is interesting to observe that when directly evaluated on a different domain that the models are trained from, we could still get non-trivial test performance. Moreover, P-T consistently outperforms RI-T even in the cross-domain cases. A more surprising observation is that when interpolating between P-T models, (instead of performance barrier) we observe performance boost in the middle of the interpolation. This suggests that all the trained P-T models on all domains are in one shared basin.

## B.8 Cross-domain weight interpolation with training on combined domains

In this section, we investigate interpolation with models that are trained on combined domains. In particular, some models are trained on a dataset formed by the union of the training set from multiple DOMAINNET domains. We tested the following scenarios:

Figure 19: Performance barrier of cross-domain interpolation. The test accuracy (left) and the cross entropy loss (right) are evaluated on the `clipart` domain. The interpolation are between models trained on `clipart` and models trained on `real`.

Figure 20: Performance barrier of cross-domain interpolation. The test accuracy (left) and the cross entropy loss (right) are evaluated on the `clipart` domain. The interpolation are between models trained on `quickdraw` and models trained on `real`.

Figure 21: Performance barrier of cross-domain interpolation. The test accuracy (left) and the cross entropy loss (right) are evaluated on the `real` domain. The interpolation are between models trained on `quickdraw` and models trained on `clipart`.

Figure 22: Performance barrier of cross-domain interpolation. The test accuracy (left) and the cross entropy loss (right) are evaluated on the `real` domain. The interpolation are between models trained on `real` and models trained on `clipart`.

Figure 23: Performance barrier of cross-domain interpolation. The test accuracy (left) and the cross entropy loss (right) are evaluated on the `real` domain. The interpolation are between models trained on `real` and models trained on `quickdraw`.

Figure 24: Performance barrier of cross-domain interpolation with training on combined domains. The test accuracy (left) and the cross entropy loss (right) are evaluated on the `real` domain. The interpolation are between models trained on `real+clipart` and models trained on `clipart`.

Figure 25: Performance barrier of cross-domain interpolation with training on combined domains. The test accuracy (left) and the cross entropy loss (right) are evaluated on the `real` domain. The interpolation are between models trained on `real+quickdraw` and models trained on `quickdraw`.

Figure 26: Performance barrier of cross-domain interpolation with training on combined domains. The test accuracy (left) and the cross entropy loss (right) are evaluated on the `real` domain. The interpolation are between models trained on `real+quickdraw` and models trained on `real`.

Figure 27: Performance barrier of cross-domain interpolation with training on combined domains. The test accuracy (left) and the cross entropy loss (right) are evaluated on the `real` domain. The interpolation are between models trained on `clipart+quickdraw` and models trained on `clipart`.

Figure 28: Performance barrier of cross-domain interpolation with training on combined domains. The test accuracy (left) and the cross entropy loss (right) are evaluated on the `clipart` domain. The interpolation are between models trained on `real+clipart` and models trained on `real`.

Figure 29: Performance barrier of cross-domain interpolation with training on combined domains. The test accuracy (left) and the cross entropy loss (right) are evaluated on the `clipart` domain. The interpolation are between models trained on `real+quickdraw` and models trained on `quickdraw`.

| Results | Evaluated on | Training data for Model 1 | Training data for Model 2 |
|---------|--------------|---------------------------|---------------------------|
| Figure 24 | `real` | `real+clipart` | `clipart` |
| Figure 25 | `real` | `real+quickdraw` | `quickdraw` |
| Figure 26 | `real` | `real+quickdraw` | `real` |
| Figure 27 | `real` | `clipart+quickdraw` | `clipart` |
| Figure 28 | `clipart` | `real+clipart` | `real` |
| Figure 29 | `clipart` | `real+quickdraw` | `quickdraw` |

## B.9 Additional criticality plots

Figure 30 in shows the criticality analysis for Conv1 module of the ResNet-50 using training data or test data or generalization gap. As we see, all of them can be used interchangeably for the analysis. The accompanying file 'criticality-plots-chexpert.pdf' includes the figures from main text along with many more such plots for different layers of ResNet-50.

## Appendix C: Spectrum of weight matrices

We recover the spectrum of every module using the algorithm derived in [Sedghi et al., 2019] and we also look at the spectrum for the whole network in different cases of training. Sedghi et al. [2019] proposes an exact and efficient method for finding all singular values corresponidng to the convolution layers with a simple two-lines of NumPy which essentially first takes $2D$-FFT of the kernel and then takes the union of singular values for different blocks of the result to find all singular values. Figure 31a shows the spectrum of the whole network for different models for CHEXPERT domain. The plots for other domains and for individual modules are shown in the Supplementary material. We note that for individual modules as well as the whole network, RI-T is more concentrated towards zero. In other words, it has a higher density in smaller singular values. This can be seen in Figure 31a, 31b. In order to depict this easier, we sketch the number of singular values smaller than some threshold vs the value of the threshold in Figure 33. Intuitively, among two models that can classify with certain margin, which translates to having low cross-entropy loss, then we can look at concentration of spectrum and concentration towards low values shows less confidence. More mathematically speaking, the confident model requires a lower-rank to get $\epsilon$-approximation of the function and therefore, has lower capacity. Intuitively, distribution around small singular values is a hint of model uncertainty. When starting from pre-trained network, the model is pointing strongly into directions that have signals about the data. Given that RI-T does not start with strong signals about the data, it

Figure 30: Module Criticality plots for Conv1 module. x-axis shows the distance between initial and optimal $\theta$, where $x = 0$ maps to initial value of $\theta$. y-axis shows the variance of Gaussian noise added to $\theta$. The four subplots refer to the four paths one can use to measure criticality. All of which provide good insight into this phenomenon. Each subplot has 3 rows corresponding to train error, test error and generalization error. Heat map is used so that the colors reflect the value of the measure under consideration.

(a) Histogram of spectrum of the whole model      (b) Lower part of spectrum

Figure 31: Spectrum of the whole network, CHEXPERT.

Figure 32: Spectrum of the whole model for target domain Clipart

(a) CHEXPERT

(b) clipart

Figure 33: Count of singular values smaller than a threshold

finds other explanations compared to P-T and hence ends up in a different basin of loss landscape, which from a probabilistic perspective is more concentrated towards smaller singular values.

Figure 32 shows the spectrum of the whole network for target domain clipart. The accompanying files 'spectrum-plots-chexpert.pdf', 'spectrum-plots-clipart.pdf' in the Supplementary material folder include the spectrum for each module of ResNet-50 as well as the whole spectrum for target domain CHEXPERT, clipart respectively.

There is a vast literature in analyzing generalization performance of DNNs by considering the ratio of Frobenius norm to spectral norm for different layers [Bartlett et al., 2017, Neyshabur et al., 2018]. They prove an upper bound on generalization error in the form of $O\left(1/\gamma, B, d, \Pi_{i\in l}\|\theta_i\|_2 \sum_{i\in l} \frac{\|\theta_i\|_F}{\|\theta_i\|_2}\right)$ where $\gamma, B, d, l$ refer to margin, norm of input, dimension of the input, depth of the network and $\Theta_i$s refer to module weights and Frobenius and spectral norm are shown with $\|\cdot\|_F, \|\cdot\|_2$. For details, see [Neyshabur et al., 2018]. The product of spectral norm can be considered as a constant times the margin, B,d are the same in the two networks. Therefore, we compare the term $\sum_{i\in d} \frac{\|W_i\|_F}{\|W_i\|_2}$ for two networks. Calculating this value shows bigger generalization bound for RI-T and hence predicts worse generalization performance.

## Footnotes

[5] `https://quickdraw.withgoogle.com/data`


[Supplementary Material 2]

# Supplementary Material for
# 'What is being transferred in transfer learning?'
# Spectrum plots for Clipart

Behnam Neyshabur*
Google
neyshabur@google.com

Hanie Sedghi*
Google Brain
hsedghi@google.com

Chiyuan Zhang*
Google Brain
chiyuan@google.com

_______________________

*Equal contribution. Authors ordered randomly.

**Figure 1: Spectrum for the whole network**

**Figure 2: Spectrum Conv1**

**Figure 3: Spectrum Layer1.0 Conv1**

**Figure 4: Spectrum Layer1.0 Conv2**

**Figure 5: Spectrum Layer1.0 Conv3**

**Figure 6: Spectrum Layer1.0 Downsample**

**Figure 7: Spectrum Layer1.1 Conv1**

**Figure 8: Spectrum Layer1.1 Conv2**

**Figure 9: Spectrum Layer1.1 Conv3**

**Figure 10: Spectrum Layer1.2 Conv1**

Figure 11: Spectrum Layer1.2 Conv2

Figure 12: Spectrum Layer1.2 Conv3

**Figure 13: Spectrum Layer2.0 Conv1**

**Figure 14: Spectrum Layer2.0 Conv2**

**Figure 15: Spectrum Layer2.0 Conv3**

**Figure 16: Spectrum Layer2.0 Downsample**

**Figure 17: Spectrum Layer2.1 Conv1**

**Figure 18: Spectrum Layer2.1 Conv2**

**Figure 19: Spectrum Layer2.1 Conv3**

**Figure 20: Spectrum Layer2.2 Conv1**

**Figure 21: Spectrum Layer2.2 Conv2**

**Figure 22: Spectrum Layer2.2 Conv3**

**Figure 23: Spectrum Layer2.3 Conv1**

**Figure 24: Spectrum Layer2.3 Conv2**

**Figure 25: Spectrum Layer2.3 Conv3**

**Figure 26: Spectrum Layer3.0 Conv1**

**Figure 27: Spectrum Layer3.0 Conv2**

**Figure 28: Spectrum Layer3.0 Conv3**

**Figure 29: Spectrum Layer3.0 Downsample**

**Figure 30: Spectrum Layer3.1 Conv1**

**Figure 31: Spectrum Layer3.1 Conv2**

**Figure 32: Spectrum Layer3.1 Conv3**

**Figure 33: Spectrum Layer3.2 Conv1**

**Figure 34: Spectrum Layer3.2 Conv2**

**Figure 35: Spectrum Layer3.2 Conv3**

**Figure 36: Spectrum Layer3.3 Conv1**

**Figure 37: Spectrum Layer3.3 Conv2**

**Figure 38: Spectrum Layer3.3 Conv3**

**Figure 39: Spectrum Layer3.4 Conv1**

**Figure 40: Spectrum Layer3.4 Conv2**

**Figure 41: Spectrum Layer3.4 Conv3**

**Figure 42: Spectrum Layer3.5 Conv1**

**Figure 43: Spectrum Layer3.5 Conv2**

**Figure 44: Spectrum Layer3.5 Conv3**

**Figure 45: Spectrum Layer4.0 Conv1**

**Figure 46: Spectrum Layer4.0 Conv2**

**Figure 47: Spectrum Layer4.0 Conv3**

**Figure 48: Spectrum Layer4.0 Downsample**

**Figure 49: Spectrum Layer4.1 Conv1**

**Figure 50: Spectrum Layer4.1 Conv2**

**Figure 51: Spectrum Layer4.1 Conv3**

**Figure 52: Spectrum Layer4.2 Conv1**

**Figure 53: Spectrum Layer4.2 Conv2**

**Figure 54: Spectrum Layer4.2 Conv3**



[Supplementary Material 3]

# Supplementary Material for
# 'What is being transferred in transfer learning?'
# Spectrum plots for ChexPert

Behnam Neyshabur[*]
Google
neyshabur@google.com

Hanie Sedghi[*]
Google Brain
hsedghi@google.com

Chiyuan Zhang[*]
Google Brain
chiyuan@google.com

[*]Equal contribution. Authors ordered randomly.

Figure 1: Spectrum for the whole network

Figure 2: Spectrum Conv1

**Figure 3: Spectrum Layer1.0 Conv1**

**Figure 4: Spectrum Layer1.0 Conv2**

**Figure 5: Spectrum Layer1.0 Conv3**

**Figure 6: Spectrum Layer1.0 Downsample**

**Figure 7: Spectrum Layer1.1 Conv1**

**Figure 8: Spectrum Layer1.1 Conv2**

**Figure 9: Spectrum Layer1.1 Conv3**

**Figure 10: Spectrum Layer1.2 Conv1**

**Figure 11: Spectrum Layer1.2 Conv2**

**Figure 12: Spectrum Layer1.2 Conv3**

**Figure 13: Spectrum Layer2.0 Conv1**

**Figure 14: Spectrum Layer2.0 Conv2**

**Figure 15: Spectrum Layer2.0 Conv3**

**Figure 16: Spectrum Layer2.0 Downsample**

**Figure 17: Spectrum Layer2.1 Conv1**

**Figure 18: Spectrum Layer2.1 Conv2**

**Figure 19: Spectrum Layer2.1 Conv3**

**Figure 20: Spectrum Layer2.2 Conv1**

**Figure 21: Spectrum Layer2.2 Conv2**

**Figure 22: Spectrum Layer2.2 Conv3**

**Figure 23: Spectrum Layer2.3 Conv1**

**Figure 24: Spectrum Layer2.3 Conv2**

**Figure 25: Spectrum Layer2.3 Conv3**

**Figure 26: Spectrum Layer3.0 Conv1**

**Figure 27: Spectrum Layer3.0 Conv2**

**Figure 28: Spectrum Layer3.0 Conv3**

**Figure 29: Spectrum Layer3.0 Downsample**

**Figure 30: Spectrum Layer3.1 Conv1**

**Figure 31: Spectrum Layer3.1 Conv2**

**Figure 32: Spectrum Layer3.1 Conv3**

**Figure 33: Spectrum Layer3.2 Conv1**

**Figure 34: Spectrum Layer3.2 Conv2**

**Figure 35: Spectrum Layer3.2 Conv3**

**Figure 36: Spectrum Layer3.3 Conv1**

**Figure 37: Spectrum Layer3.3 Conv2**

**Figure 38: Spectrum Layer3.3 Conv3**

**Figure 39: Spectrum Layer3.4 Conv1**

**Figure 40: Spectrum Layer3.4 Conv2**

**Figure 41: Spectrum Layer3.4 Conv3**

**Figure 42: Spectrum Layer3.5 Conv1**

**Figure 43: Spectrum Layer3.5 Conv2**

**Figure 44: Spectrum Layer3.5 Conv3**

**Figure 45: Spectrum Layer4.0 Conv1**

**Figure 46: Spectrum Layer4.0 Conv2**

**Figure 47: Spectrum Layer4.0 Conv3**

**Figure 48: Spectrum Layer4.0 Downsample**

**Figure 49: Spectrum Layer4.1 Conv1**

**Figure 50: Spectrum Layer4.1 Conv2**

**Figure 51: Spectrum Layer4.1 Conv3**

**Figure 52: Spectrum Layer4.2 Conv1**

**Figure 53: Spectrum Layer4.2 Conv2**

**Figure 54: Spectrum Layer4.2 Conv3**



[Supplementary Material 4 · criticality-plots.pdf]

# Supplementary Material for
# 'What is being transferred in transfer learning?'
# Criticality plots for Chexpert

Behnam Neyshabur*
Google
neyshabur@google.com

Hanie Sedghi*
Google Brain
hsedghi@google.com

Chiyuan Zhang*
Google Brain
chiyuan@google.com

(a) P-T, direct path

(b) RI-T, direct path

(c) P-T, optimization path

(d) RI-T, optimization path

Figure 1: **Module Criticality Conv1**

(a) P-T, direct path

(b) RI-T, direct path

(c) P-T, optimization path

(d) RI-T, optimization path

Figure 2: **Module Criticality FC**

(a) P-T, direct path

(b) RI-T, direct path

(c) P-T, optimization path

(d) RI-T, optimization path

Figure 3: **Module Criticality Layer1**

(a) P-T, direct path

(b) RI-T, direct path

(c) P-T, optimization path

(d) RI-T, optimization path

Figure 4: **Module Criticality Layer2**

(a) P-T, direct path

(b) RI-T, direct path

(c) P-T, optimization path

(d) RI-T, optimization path

Figure 5: **Module Criticality Layer3.0**

Figure 6: **Module Criticality Layer3.1**

(a) P-T, direct path

(b) RI-T, direct path

(c) P-T, optimization path

(d) RI-T, optimization path

Figure 7: **Module Criticality Layer3.2**

Figure 8: **Module Criticality Layer3.3**

(a) P-T, direct path

(b) RI-T, direct path

(c) P-T, optimization path

(d) RI-T, optimization path

Figure 9: **Module Criticality Layer3.4**

(a) P-T, direct path

(b) RI-T, direct path

(c) P-T, optimization path

(d) RI-T, optimization path

Figure 10: **Module Criticality Layer3.5**

(a) P-T, direct path

(b) RI-T, direct path

(c) P-T, optimization path

(d) RI-T, optimization path

Figure 11: **Module Criticality Layer3**

(a) P-T, direct path

(b) RI-T, direct path

(c) P-T, optimization path

(d) RI-T, optimization path

Figure 12: **Module Criticality Layer4**

## Footnotes

*Equal contribution. Authors ordered randomly.