[Reviews · NeurIPS 2020]

Review 1

Summary and Contributions: In this paper, what enables a successful transfer and which part of the network is responsible for that are explored by conducting a series of experiments. To be specific, role of feature reuse, mistakes predicted by networks, feature similarity, distance in feature space, performance barriers and basins in the loss landscape, module criticality, spectrum of different modules, useful pre-trained checkpoint are explored in this paper. As a result, several intersting observations and insightful discussions are provided.

Strengths: + The paper is well written. It is enjoyable to read. + Motivations are clear, some well-designed experiments are conducted and interesting observations are provided. + A portion of analyses are insightful.

Weaknesses: Important issues: - Line 137-138: More evidences should be provided, this conclusion (higher order statistics of the data that are not ruined in the shuffling lead to the significant benefits of transfer learning, especially on optimization speed) is not well supported. -Line 179-188: Is there any evidence to support the effectiveness of the proposed measurement? In my opinion, linear interpolation of the two weights is somewhat unreasonable: it can only reflect the linear relationship between the two networks, which is not a reasonable measurement for nonlinear networks. For further improving the paper: -Line 78-79: [9] show that transfer (even between similar tasks) does not necessarily result in performance improvements. This sentence is misleading: transfer does not necessarily result in performance improvements in some different downstream tasks, while in recognition this is not the case. -Line 108-109: four target domains with decreasing visual similarities from natural images. Is there any evidence/measurement to confirm this view? -Line 150: PT --> P-T -Line 197-198: a interesting observation, but there is no analysis for that.

Correctness: I did not find any technical error in the paper.

Clarity: Motivations, ideas and experimental results are well expressed in this paper.

Relation to Prior Work: Yes

Reproducibility: Yes

Additional Feedback: After reading the rebuttal, I find that main concerns are addressed and my score is 6 now.


Review 2

Summary and Contributions: This work presents a set of tools and analysis of transfer learning algorithms. Key observations are made from the study of feature-reuse, loss landscape, module criticality and training convergence. The results are verified on transfer learning tasks across datasets of varying levels of visual similarity (ImageNet, CheXpert and DomainNet). Overall, the paper provides new insights for transfer learning.

Strengths: - Transfer learning is studied from various perspectives on real-world datasets. - The presented results are insightful. - The submission of code is helpful.

Weaknesses: The authors should address the following points to improve the submission. 1. There are several references to the appendix, including major results such as the discussion in Sec. 3.2. Thus the reader has to refer to the appendix back and forth. The authors could improve the readability of the paper by incorporating certain results from the appendix to the main paper (see additional feedback section for suggestions to improve space usage). 2. Certain claims require further clarifications: a) L151-152: The following phrase is unclear, “since P-T has strong prior, it finds it harder to adapt to the target domain”. From Fig. 2 it is clear that P-T achieves a better performance than RI-T, and as described in L132, the optimization speed on P-T is more stable than RI-T for smaller block sizes. Likewise, the performance of P-T is better under class imbalance (Appendix L445-446). This is opposite to the claim in L151-152. b) L191-192: “pre-trained weights guide the optimization to a flat basin”. The information in the figures such as Fig. 3 is insufficient to argue about the flatness of the basin. Note that the x-axis in Fig. 3 is the interpolation coefficient and thus the characteristics of the basin are not guaranteed to be observed along the direction of interpolation. For e.g., the case could be that two finetuneT models are close to each other in the parameter space (due to similar initializations) and thus there is not much variation in the loss landscape along the interpolation. Whereas a pair of randinitT could be far apart (due to different random initializations), and therefore a significant barrier is observed along the interpolation. However, since both plots are overlaid on the same graph (with the x-axis being the interpolation coefficient), it provides a false notion of the “flatness” of the basin. This clarification should be incorporated unless justified otherwise. c) Sec. 3.7 Spectrum: This section requires a brief introduction of the method [26] used to obtain the spectrum. More importantly, there are several counter-intuitive examples to the discussion in Sec. 3.7. For instance, in spectrum-plots-clipart.pdf: Fig. 17, 20, 23, 26, 29, 36, 39 (right) show more concentration towards smaller singular values after the model is trained (blue) when compared with random initialization (orange). This would imply a higher uncertainty after training (L254) which is counterintuitive. Likewise, several CheXpert plots ( Fig. 3, 4, 6, 10, 11, 13, 14, 16, 17 (right) ) show a stronger result. It is important to discuss this phenomenon. Why is the model driven towards smaller singular values during training (note that this is true for fine-tune plots as well)? Perhaps, the random initialization itself possesses larger singular values, while the training along with regularizers such as weight decay (Appendix L410) drives the model towards lower singular values. In such a scenario, the spectrum may not reflect model uncertainty (L245) and the plots for P-T and RI-T are not directly comparable. 3. Fig. 5: This figure is difficult to interpret - please label the x-axis, y-axis and the color scale. What does the x-axis refer to here? (x-axis contains fractions for Fig. 5a,c and integers for Fig. 5b,d)

Correctness: Certain claims need further justification (see weaknesses).

Clarity: The paper is well written for most of the parts. However one has to refer to the appendix files back and forth to completely understand the proposed ideas. A reorganization of certain sections is required (see additional comments).

Relation to Prior Work: The paper presents a set of analyses to study the effect of transfer learning. While some techniques are based on prior works, the authors sufficiently justify the choice of tools used for analysis.

Reproducibility: Yes

Additional Feedback: Please reorganize the sections with more discussion at appropriate places. For e.g., better space adjustment can be employed in Fig. 5 & 6; certain sentences can be paraphrased to conserve space (save lines such as L262, L75, L167 and the last line of Fig. 7 caption). Similarly, the space pertaining to Table 1 can host for instance, Table B.2 after some space adjustment (using abbreviation etc.). In Fig. 5, the sub figures for the “train” statistics can be moved to the supplementary. L74: “depends of” -> “depends on” L119: “disrupt” -> “disrupts” L247: Please add a brief description of the algorithm in [26] L292: “he” -> “the” Post Rebuttal: The manuscript seems to have been improved by incorporating most of the suggestions received in the reviews. Further, my major concerns have been addressed. Therefore, I increase my score. I would recommend the authors to incorporate the clarification regarding the "connectivity" between models, and the "concentration of spectrum" in the main text."


Review 3

Summary and Contributions: The paper presents an empirical study of transfer learning using the ImageNet pretrained weights for a number of downstream classification problems. The paper makes the following contributions: 1) an analysis of feature reuse for transferring representations, considering various pretrain and target domain distribution shifts. 2) Statistical comparisons between finetuning the pretrained model and training from scratch. 3) For transfer learning, a quantitative measure of layer criticality and an investigate into the interim checkpoints for transfer performance.

Strengths: * This work proposes a series of tools to analyze transfer learning. Such as the block shuffle to disentangle the feature reuse and high-level statistics, module criticality and feature similarity. * The discovery and findings in the paper are interesting. For example, both low-level feature reuse and high-level semantics are important for transfer learning, and that interim checkpoints do not affect significantly for the transfer performance.

Weaknesses: * For all the downstream tasks investigated, it seems all tasks benefit from pretraining. I am wondering is there a task which would suffer from pretraining? * While these analysis are interesting, it seems such analysis is of little practice usage and applications. For example, there's no straightforward to improve transfer learning based on the provided understanding of transfer learning. * A lack of considering the amount of labels in the target domain. Since the paper compares finetuning against training scratch, it would be systematically analyze

Correctness: I find no significant error claims in the paper.

Clarity: Yes, the paper is easy to follow.

Relation to Prior Work: Related works in transfer learning are properly cited in the related work section. No specific discussions are made to any work.

Reproducibility: No

Additional Feedback: While I am satisfied with the submission, the rebuttal failed to address my concerns. The practical usage is not experimentally validated in any scenario. I downgrade my score to 6.


Review 4

Summary and Contributions: The paper targets the question: what enables a successful transfer? and which parts of the network are responsible for this? To answer these, they experiment with various methods, comparing pretrained models, fine-tuning on pretrained models and models trained from scratch using random initialization.

Strengths: The provided observations are conducted on a variety of datasets.

Weaknesses: While the paper targets an important question, it lacks in novelty and did not do justice to the targeted problem. Several of the findings are either well known or are expected. E.g. the paper claims that lower layers are in-charge of feature reuse. However, this finding is not fully supported by the evidence.

Correctness: Most of the claims are supported with evidence. I have provided feedback in detailed comments.

Clarity: The paper lacks in clarity. Since the paper is based on already published methods, a few lines description of those techniques and motivation of why authors prefer a particular approach are missing from the paper.

Relation to Prior Work: Most of the work from vision is covered. However, there are a number of attempts in NLP that analyze pre-trained models before and after fine-tuning. Authors should acknowledge the work in NLP.

Reproducibility: Yes

Additional Feedback: - Role of feature reuse: by comparing fine-tuned models on real data and on data from clip art and quick draw, authors claim that feature reuse plays an important rule in transfer learning. However, it is not clear which features are reused. Secondly, it is an expected output that finetuning on related data helps more than finetuning on unrelated data. The finetuning of unrelated data can be improved given the large size of the finetuning data. Overall, I am not sure how to connect the performance as a measure of feature reuse. - Again looking at faster convergence of P-T compared to RI-T, this is quite obvious since P-T is starting from pretrained weights which are already optimized and the fine-tuning step is adapting the weights towards the target domain. On the other hand, RI-T is starting from a random place and would require a lot more time to optimize. - Results on randomly shuffling the blocks in an image: again it is understandable that shuffling the image would drop the performance and it is likely to drop more for RI than P since P is started from a well optimized point. - Authors mentioned (line 131) in the case of quick draw, some other factors are helping the downstream tasks. Do you any intuition on what are those features? Which part of the network they belong to? - It is interesting to observe that two P-Ts trained on the same data but with different initialization make similar mistakes. I am wondering what could be the reason? Is it because you are looking at the models from the last epoch. So models may start from very different points in the space but with the large size of the data and a good number of training epochs, they finally converge to approximately the same point. I remembered that ensembling two different checkpoints of a same model gives performance improvement. This might be happening since both checkpoints were far from each other in the space and they may be making different mistakes. I think checkpoint-wise comparison is an important point to understand whether the reason of two PTs making the same mistakes and also have a similar flat basin is the result of using identical large data plus a large number of epochs. - The above comment is also valid for the case of feature similarity (3.3). Looking a table 1, I am wondering if RI-T being very different from RI-T and others is because of the small size of the training data? The data is small for the model to generalize well and it may stuck in some local optimum? - One interesting observation in Table 1 is that higher layers are more different than lower layers. I understand that lower layers are similar because they are closer to input and represent input features while higher layers learn abstract features plus features more optimized towards the objective function. Since two P-Ts are using identical data and identical base model, it would be interesting to know why they are different on high layers while the loss landscape shows that these models belong to the same flat basin. - Loss landscape: did authors play with layer-wise weight difference in two models? I suspect - The paper should justfiy the choice of techniques. Why cka or singular value analysis using Sedghi et al.? I would expect to have a samll description of each method used in the paper instead of just saying, we used the method in Sedghi et al. for our analysis. Minor comments: - Why to introduce RI? it is confusing since this is not a trained model but just random weights and you are not using it in the paper. typos: line 269: 29, 30, 30 -> 29, 30, 31 line 292: he -> the

[Author Response · NeurIPS 2020]

We thank the reviewers for their detailed comments and questions. Typos are fixed.

**R1** **Line 137-138**: In these experiments we change the size of the shuffled blocks all the way to 1 and even try shuffling
the channels of the input. Therefore, the only object that is preserved here is the set of all pixel values which can be
looked at as a histogram/distribution and be fully characterized by its moments. We have now clarified this in the paper.
**Line 179-188**: It has been observed in the literature that *any* two neural network minimizers can be connected via a
non-linear low-loss path. In contrast, as mentioned by the reviewer, due to non-linear nature of neural nets one would
not expect the model obtained by the linear interpolation of two neural nets in the parameter space to perform well, and
that is precisely why we find this phenomenon interesting and investigate it. Note that insights from this observation can
be used for improving ensemble methods. **Line 108-109**: One can come up with quantitative measure by comparing
domains higher level representations by passing them through a network but here what we meant was looking at data
itself, and we have included some samples in the paper. Real includes real images, while Clipart is a cartoon version,
both have colors and texture. On the other hand ChexPert is black and white and has texture while Quickdraw has no
color or texture. In light of addressing your main concerns, we respectfully ask you to consider accepting the paper.

**R2** **Back n forth to Appendix**: Thank you for your suggestions. We have now moved the distance in feature space
results to the main part of the paper and moved the 'train' part of figure 5 to the Appendix. **L151-152**: We mean
ambiguous data points in the target domain. We will revise the text to clarify this. **L191-192**: Two P-T models in the
same basin are not trivially (approximately) equal, as higher layers compute quite different features (Table 1). We
agree that using interpolation coefficient as x-axis does not tell the whole story. But with ReLU you can artificially
scale up weight magnitudes and distances. It's the connectivity rather than the actual distance that is important here.
Regarding RI-T having different random seeds, we added experiments of different RI-T models from *the same* random
initialization values. They still converge to *different disconnected* basins. We also added extrapolation (with coefficient
in [-1, 2]) to our connectivity analysis. **Spectrum section**: Thank you for your question, what we meant is among two
models that can classify with certain margin, which translates to having low cross-entropy loss, then we can look at
concentration of spectrum and concentration towards low values shows less confidence. More mathematically speaking,
the confident model requires a lower-rank to get $\epsilon$-approximation of the function and therefore, has lower capacity.
However, the model at init (either random or pre-train) has higher loss and is not fitted to the data.

**R3** **Examples of negative transfer?** Pre-training with random labels *can* hurt transfer learning [Maennel et al 2020].
**Practical usage**: Our findings on basin in the loss landscape can be used to improve ensemble methods. Our observation
of higher order statistics improving training speed could lead to better network initialization methods.

**R4** **Novelty**: The block-shuffling experiments and the linear interpolation of two neural networks experiments are
not trivial and it was unclear what to expect beforehand. They were not done before either. Moreover, experiment
outcomes matching the intuitions does not necessarily renders the experiments uninteresting or unimportant. **Claims**:
We clarify that we mention in the paper "lower layers are in charge of more general features" (comparing to more
class-specific features in higher layers) and we do not claim they are in charge of feature reuse. (line 240-242) **Explain**
**methods used**: CKA[Kornblith et al] is the latest work on estimating feature similarity with superior performance
over earlier works. The algorithm in [Sedghi et al] is the one that provides correct singular values for convolutions
and has been used for various applications. We cited the works and referred the reader to original papers for details
for space constraints. We have now added details of both methods to the Appendix. **Related work in NLP**: We have
now added [Brown+2020, Devlin+2018, Tamkin+2020, Raffel+2019, Liu+2015, Radford+2019, Roberts+2020] to the
related works on transfer learning in NLP. **Difference between RI-T & P-T**: The reviewer has brought up this point
multiple times that better performance of P-T, P-T's being in the same basin, and the results of shuffling experiments is
*expected* due to the fact that it starts from an optimized point. We need to emphasize that the pre-training model has
been optimized to fit a *different* dataset (e.g. ImageNet). It would not necessarily converge faster or perform better in
downstream tasks (esp. with shuffled blocks) simply because it is 'already optimized'. In fact, if the outcome is the
opposite, we could perfectly justify it by saying that it is trapped in a local optima because it is 'already optimized'
(to fit a different task). Moreover, optimization speed is affected differently from accuracy. **Two P-T's in the same**
**basin being a result of large number of epochs**: We have performed many experiments and also results from Section
4 show that once the models pass the initial finetuning stage, they belong to the same basin and a large number of
epochs is not needed. We also added new experiments of two RI-T models from *the same* initializations, and unlike
P-T, they do *not* converge to the same basin even after many epochs. **Higher layers being less similar compared to**
**lower layers**: The two P-T models are in the same basin but are not identical functions. Stochasticity of SGD leads to
non-identical models. Lower layers are the foundation of the function and higher layers depend on lower layers. If
the lower layer feature is more robust there are many solutions at higher layers and each of them can perform well.
Moreover, in a ResNet model you may not be using the weights in the higher layers and may be using skip connection.
Nevertheless, P-T's are more similar compared to other models (Table 1). In light of addressing your main concerns, we
respectfully ask you to consider accepting the paper.

[Meta-Review · NeurIPS 2020]

This paper provides experimental results and analyses from multiple perspectives for revealing what enables a successful transfer and which part of the network is responsible for that. Reviewers and AC unanimously agree that this paper is well written, proposes new tools for understanding transfer learning and provides novel and important insights. The rebuttal addresses most of the concerns raised by the reviewers. After rebuttal, a reviewer still concerns about the practical value of the understanding since it does not imply a real technique to promote transfer performance. The paper is recommended for acceptance. Please make sure to incorporate the rebuttal material into the camera-ready version.